# Seroprevalence of *Toxoplasma gondii* and Associated Risk Factors in Pregnant Women in Araçatuba, São Paulo, Brazil: A Multi-Level Analysis

**DOI:** 10.3390/microorganisms12112183

**Published:** 2024-10-30

**Authors:** Tatiani da Silva Palhota Lozano, Aline Benitez, Joice Cristina dos Santos, Italmar Teodorico Navarro, Walter Bertequini Nagata, Michel dos Santos Pinto, Jancarlo Ferreira Gomes, Gabriele Zaine Teixeira Debortoli, Thaís Rabelo Santos-Doni, Katia Denise Saraiva Bresciani

**Affiliations:** 1Faculdade de Medicina Veterinária, Universidade Estadual Paulista (UNESP), Araçatuba 16050-680, SP, Brazil; tatiaenf@yahoo.com.br (T.d.S.P.L.); 31benitez.aline@gmail.com (A.B.); joicesantos@prof.educacao.sp.gov.br (J.C.d.S.); walter.bn@hotmail.com (W.B.N.); ms.pinto@unesp.br (M.d.S.P.); 2Departamento de Medicina Veterinária Preventiva, Universidade Estadual de Londrina (UEL), Londrina 86057-970, PR, Brazil; italmar@uel.br; 3Faculdade de Ciências Médicas, Instituto de Computação, Universidade Estadual de Campinas, Campinas 13083-887, SP, Brazil; jgomes@ic.unicamp.br; 4Instituto de Ciências Agrárias (ICA), Universidade Federal dos Vales do Jequitinhonha e Mucuri (UFVJM), Avenida Universitária, Unaí 38610-000, MG, Brazil; gabriele.zaine@ufvjm.edu.br

**Keywords:** toxoplasmosis, epidemiology, prenatal care, congenital toxoplasmosis, public health, health education

## Abstract

This study assessed the seroprevalence of *Toxoplasma gondii* and risk factors among 428 pregnant women attending Basic Health Units (BHUs) in Araçatuba, São Paulo, Brazil. The seroprevalence was 55.14%, indicating high exposure to the parasite in this population. Using a multi-level logistic regression model, this study analyzed these predictors to determine their association with a higher seropositivity rate, with BHUs included as a random factor. Predictors associated with higher seropositivity included older age (36–45 years), with a 71.64% prevalence in this group, and multiparity (61.65%). Women with lower educational levels were also more likely to be infected, with 59.46% seropositivity recorded among those who had only completed elementary school. Despite identifying several risk factors, no significant correlation was found between undercooked meat consumption or contact with soil and infection. These findings highlight the need for targeted public health interventions, particularly for educating high-risk groups about toxoplasmosis prevention, such as safe food handling and avoiding raw dairy products. Additionally, BHUs play a critical role in early detection and prevention. These units are important for providing healthcare access and preventive education for vulnerable populations. Given the high seroprevalence, this study underscores the urgency of implementing prenatal screening and educational programs to reduce the risks of congenital toxoplasmosis in this region.

## 1. Introduction

Toxoplasmosis is a globally distributed disease caused by the protozoan *Toxoplasma gondii* [1], which has a heteroxene cycle and reproduces sexually in the intestines of felids (definitive hosts) and asexually in all warm-blooded animals, such as birds and mammals, which act as intermediate hosts [2].

Ingestion of *T. gondii* through undercooked meat and contaminated food [3,4,5,6], contact with the soil during gardening or with a litter box where felines defecate [7,8,9], owning a pet cat [10], consuming unpasteurized cow’s or goat’s milk [11,12], eating raw fruits and vegetables [13,14], educational level and occupation [15,16], age [17,18], being foreign-born/a minority [19,20,21], drinking unregulated water [22,23,24], and parity [19,25] are considered important risk factors for infection in pregnant women. In addition, other infectious routes involve organ transplantation and blood transfusion [26,27,28].

The transplacental route can lead to serious damage to pregnancy such as premature birth, abortion, stillbirth, fetal malformation, and infant morbidity and mortality [29,30].

Approximately one-third of the world’s population is exposed to the parasite, with greater severity in tropical regions [31,32].

In Brazil, up to 50% of elementary school children and 50–80% of women of child-bearing age have antibodies to *T. gondii*. This occurrence can be justified by several factors that contribute to its transmission, such as favorable climatic conditions, environmental contamination, the large number of free-living intermediate hosts, basic sanitation deficit, and deficient implementation of the laws that govern the responsibilities and duties of the state and pet guardians [33,34].

The prevalence of congenital toxoplasmosis is influenced by the social, economic, and regional factors; eating habits; maternal age; and geographic location of each population. Such factors must, therefore, be identified to assist with the adoption of appropriate disease prevention strategies for a given group of pregnant women. In this work, the factors associated with the occurrence of seroprevalence of *T. gondii* infection in pregnant women attending the main Basic Health Units (BHU) in Araçatuba, São Paulo, Brazil, were investigated.

## 2. Materials and Methods

### 2.1. Ethical Statement

All procedures were carried out after approval by the Ethics Committee for Research with Human Beings, São Paulo State University (UNESP, N°. 2,625,160).

### 2.2. Study Design and Serum Samples

A cross-sectional study was conducted from May 2018 to March 2019 in Araçatuba, a municipality located in the northwest of the State of São Paulo (21°12′32″ S, 50°25′58″ W), Brazil, at 380 m above sea level, which has a semi-humid tropical climate and receives water from the Tietê and Ribeirão Baguaçu rivers. In 2019, the population was estimated at 195,874 inhabitants, with 29,149 women being of reproductive age and infant mortality being 11.02 deaths per thousand live births. Still, in 2019, the HDI was 0.788 and the Gini index was 0.515; the latter measure represents the concentration of income among social strata, where 0 corresponds to complete equality and 1 corresponds to maximum inequality [35].

This investigation was conducted in partnership with the Municipal Health Secretariat (MHS) to obtain data from pregnant women assisted in four BHUs.

In the municipality, primary healthcare is made up of 19 service units: 4 are distributed in the rural area and 15 in the urban area. Of these, four units located in the peripheral regions of the city were selected for this investigation because they had the highest number of prenatal consultations in the municipality.

The sample size was calculated using the Epi Info Version 7.2.2.6, Software (CDC, Atlanta, GA, USA) by using a 95% confidence level and 5% precision, with an expected positivity of 50%; maintaining the statistical error under 1% [36]; and studying a total population of 195,874, resulting in a minimal sample size of 384 women. To encompass a large number of samples, a total of 428 women’s blood samples were collected and analyzed.

Georeferencing (Figure 1) was carried out with a GPS device (GPS Garmin eTrex^®^ 30, Olathe, TX, USA). The data were then transported to the QGIS desktop (3.26.3) program [37], where they were placed on the Araçatuba/São Paulo map. The database modeling and chart plotting stages were carried out at the Institute of Agrarian Sciences, Federal University of the Jequitinhonha and Mucuri Valleys.

### 2.3. Serum Samples

The immunological profile of antibodies against *T. gondii* through the chemiluminescence test was consulted in the medical records.

Chemiluminescence technology was used for the quantitative determination of specific anti-*T. gondii* IgG antibodies (LIAISON^®^, DiaSorin S.p.A., Salluggia, Italy). Samples were considered IgG-positive when the antibody concentration was greater than or equal to 8.8 UI/mL, non-reactive when the concentration was less than 7.2 UI/mL, and undetermined when the concentration was between 7.2 and 8.8 UI/mL.

### 2.4. Questionnaire

All pregnant women attending the four most important BHUs (BHU A, BHU B, BHU C, and BHU D), in terms of number of visits in the city, during the period, were invited to participate in this research at the time of the prenatal consultation. They were informed about the reason for the research, and those who consented to participate in the research had their medical records consulted and answered an interview guided by a data collection instrument with questions related to the epidemiology of gestational toxoplasmosis, such as the following: undercooked meat (Yes or No); manipulate soil or sand (Yes or No); eat raw vegetables and greens (Yes or No); presence of a vegetable garden at home (Yes or No); owning a pet cat (Yes or No); ingestion of unpasteurized cow’s or goat’s milk (Yes or No); gestation (1 or >1); age range (15–24 or 25–35 or 36–45); marital status (Single or Married or Other); level of education (Complete Elementary or Complete Medium or Graduated); minimum wage (≤1, 1 > 2, or >3); BHU (A, B, C, or D).

This instrument was applied by nurses from each unit and to 428 pregnant women. Subsequently, all participants received preventive guidance on gestational and congenital toxoplasmoses and their respective health problems.

### 2.5. Statistical Analyses

Statistical tests were performed using STATA/SE, Version 16.1, Software (Stata Corp LP, College Station, TX, USA).

The creation of the database and analysis of the variables were performed using the Epi Info 2007 statistical package, version 7.2.2.6, Centers for Disease Control and Prevention, Atlanta, United States (CDC). Quantitative variables were expressed as means and medians, and qualitative variables were classified and expressed as proportions. Pearson’s Chi-square (χ2) or Fisher’s exact test was executed to evaluate the differences among groups.

For inferential statistics, the seroprevalence of *T. gondii* infection was considered as the dependent variable, and other factors were considered as the explanatory variables.

To investigate the independent risk factors of each explanatory variable, all variables that showed a *p* value of ≤ 0.20 in the univariate multi-level logistic regression [38,39,40], using BHU as a random-effect variable, were offered to the multi-level logistic regression as suggested by Hutchinson et al. [41]. The examined variables were included as fixed factors in the models with BHU included as a random factor.

It is advised to use an initial screening *p* value cut-off point of 0.20, as more traditional levels such as 0.05 can fail to recognize variables known to be important. The occurrence probability ratio (odds ratio, OR) and the corresponding 95% CI were calculated by using univariate and multiple multi-level logistic regression. A *p* value of <0.05 was considered as the level of statistical significance for all tests.

## 3. Results

The study included 428 pregnant women, distributed in the four BHUs: 95 in unit A, 72 in unit B, 88 in unit C, and 173 in unit D. In 236 (55.14%) medical records, the presence of IgG antibodies against *T. gondii* was recorded (Table 1).

Table 2 shows the prevalence of IgG antibodies against *T. gondii* in pregnant women who attended Basic Health Units (BHU) in Araçatuba, SP, categorized by individual risk factors such as age group, marital status, and number of pregnancies. Data were stratified across four health units (A, B, C, and D), and the prevalence of positive cases varied according to these variables. In general, older pregnant women (36–45 years) and those with more than one pregnancy (>1) had a higher prevalence of IgG seropositivity. At BHU B, seroprevalence varied significantly between age groups (*p* = 0.005), while at BHU D, women aged 15–24 showed a lower prevalence of IgG positivity (*p* = 0.02). Marital status also influenced the results, with married women showing higher seropositivity in several health units, particularly at BHU B (*p* = 0.04). Women with multiple pregnancies showed higher seropositivity at BHU B (*p* < 0.001) and BHU D (*p* = 0.001).

Table 3 presents a univariate multi-level logistic regression evaluating risk factors associated with the presence of anti-*T. gondii* antibodies among pregnant women in Araçatuba, SP. Age was a significant predictor, with women aged 36–45 having the highest likelihood of IgG positivity (OR = 3.00, CI95%: 1.62–5.54, *p* = 0.0006), and those aged 25–35 also showing increased risk (OR = 1.60, CI95%: 1.05–2.42, *p* = 0.04). Marital status also played a role, with married women exhibiting a higher odds ratio compared to single women (OR = 1.85, CI95%: 1.14–3.01, *p* = 0.01). Pregnancy history significantly influenced seropositivity, as women with more than one pregnancy had a higher risk (OR = 2.01, CI95%: 1.35–2.99, *p* = 0.001). Other notable factors include education, where women with elementary education were at higher risk (OR = 1.47, CI95%: 1.04–2.08, *p* = 0.029), and minimum wage, where those earning ≤1 minimum wage were more likely to be IgG-positive (OR = 1.62, CI95%: 1.00–2.63, *p* = 0.049). Additionally, ingestion of unpasteurized cow’s or goat’s milk showed a borderline significance (OR = 1.71, CI95%: 0.97–3.01, *p* = 0.08).

A mixed-effect logistic regression model (Table 4) was used to investigate factors associated with positive toxoplasmosis status, controlling for the random effects of the healthcare facility (BHU). The model included age groups, marital status, income, number of pregnancies, and education levels as predictor variables.

The analysis revealed that age was a significant predictor of toxoplasmosis status. Individuals aged 36–45 (OR = 2.82, 95% CI: 1.38–5.78, *p* = 0.005) were 2.82 times more likely to test positive for toxoplasmosis compared to the reference group. The odds of testing positive were also elevated for individuals aged 25–35 (OR = 1.60, 95% CI: 0.97–2.64, *p* = 0.063), though this effect was marginally significant.

Marital status and income did not show a statistically significant association with toxoplasmosis positivity. Married individuals (OR = 1.13, 95% CI: 0.63–2.03, *p* = 0.693) and single individuals (OR = 0.71, 95% CI: 0.35–1.42, *p* = 0.326) had similar odds of testing positive. Similarly, income groups (≤1: OR = 2.10, 95% CI: 0.73–6.05, *p* = 0.171; 1 > 2: OR = 1.37, 95% CI: 0.63–2.98, *p* = 0.427) did not significantly predict toxoplasmosis status.

Although the number of pregnancies was positively associated with toxoplasmosis positivity (OR = 1.48, 95% CI: 0.91–2.39, *p* = 0.112), this effect was not statistically significant.

Education level also played an important role. Participants with complete elementary had a significantly higher likelihood of testing positive for toxoplasmosis (OR = 2.58, 95% CI: 1.07–6.19, *p* = 0.034), as did those with medium-level education (OR = 2.59, 95% CI: 1.17–5.74, *p* = 0.019), compared to individuals with higher education (reference group).

The baseline odds of testing positive, conditional on the random effects of the healthcare facility, were low (OR = 0.23, 95% CI: 0.07–0.78, *p* = 0.019). Variance in toxoplasmosis positivity across healthcare facilities was accounted for by a random intercept for BHU, with a variance estimate of 0.25 (95% CI: 0.05–1.32).

Overall, the inclusion of the random effect significantly improved the model’s fit (*p* < 0.001), confirming that facility-level differences contributed to variations in toxoplasmosis outcomes.

The random effects of the BHU indicated significant variations in the probability of *T. gondii* seropositivity. BHU 3 showed the largest positive effect (0.07), suggesting a higher likelihood of seropositivity for *T. gondii* among pregnant women attending this unit. In contrast, BHU 4 exhibited the largest negative effect (−0.60), indicating a lower probability of seropositivity compared to the global average of the model. BHU 2 showed a slightly positive effect (0.08), while BHU 1 had a moderately negative effect (−0.18). These variations among BHUs may reflect differences in the populations served or the quality and coverage of health services offered at each unit, highlighting the need for further investigation to understand the observed disparities.

## 4. Discussion

This study showed a seroprevalence of 55.14% (236/428) for *T. gondii* IgG among pregnant women in Araçatuba, São Paulo, Brazil, consistent with findings from other regions of the country [42]. The obtained value agrees with that found for other regions of the country, Martinez et al. [43] reported a 56.2% seroprevalence among pregnant women with chronic *T. gondii* infection who took part in the investigation in Nazaré and Aratuípe, Bahia. Similarly, Antinarelli et al. [44] found that 44.4% of pregnant women in Juiz de Fora, Minas Gerais, were seropositive for *T. gondii* IgG antibodies. In Presidente Prudente, São Paulo, Pereira et al. [45] observed a lower seroprevalence of 24.6% (69/280) for *T. gondii* IgG among pregnant women. A higher prevalence was reported in Fortaleza, Ceará, where 68.6% of pregnant women attending a public tertiary care hospital were seropositive for *T. gondii* IgG [46].

The seroprevalence observed in this study is significantly higher compared to other regions globally. In Namwala, Zambia, the seroprevalence of *T. gondii* IgG among pregnant women was only 4.2%, indicating a low endemic presence of the infection in that population [47]. Similarly, a study in Aguascalientes City, Mexico, reported a seroprevalence of 6.2% among 338 pregnant women, which is also significantly lower than the prevalence observed in Araçatuba [48]. These differences suggest that regional variations in environmental conditions, dietary habits, and public health measures may play a significant role in influencing the prevalence of *T. gondii* infection.

In Brazil, the tropical climate, combined with deficient sanitation and cultural practices related to food handling, creates favorable conditions for *T. gondii* survival [49]. Factors such as high humidity and contact with oocyst-contaminated environments are well-established risks, especially in rural regions where animal and human contact are frequent.

Seropositivity was highest among women aged 36 to 45 years, reaching 71.64%, which was three times higher compared to younger women; similar results were reported by Radoi et al. [50]. This finding is consistent with studies showing that exposure to the parasite increases over time, where older individuals are at higher risk due to repeated contact with contaminated food or soil [51,52]. A similar trend was observed in other tropical regions, such as Fortaleza, where the prevalence was highest among women under 25 years of age [53]. This may be related to the increased duration of exposure over a lifetime, which raises the chances of infection. Additionally, women with more than one pregnancy (multiparous) also had higher seropositivity rates, reflecting a similar pattern observed in studies like those of Hassamen et al. [11] and Lee, Lee, Tsai, Chu, Huang, Cheng, You, Huang, Lan, and Hsu [25], which highlight that multiparous women are more exposed to environmental and dietary risks that facilitate transmission.

This study found that women with more than one pregnancy were twice as likely to be seropositive for *T. gondii* (OR = 2.01). This is supported by research suggesting that multiparous women face higher exposure due to increased responsibilities and environmental risks associated with child-rearing, particularly in low-income settings [54].

Women with elementary education were 1.47 times more likely to be infected. This is consistent with studies showing that lower educational attainment correlates with limited access to preventive information, particularly in food safety and sanitation [11,55]. These women may not be fully aware of the dangers of raw or undercooked meat, a well-documented transmission route in Brazil and elsewhere [44].

The consumption of unpasteurized milk was another identified risk, with an odds ratio of 1.71. This finding is supported by Hassanen, Makau, Afifi, Al-Jabr, Abdulrahman Alshahrani, Saif, Anter, El-Neshwy, Ibrahim, and Abou Elez [11], Sohn-Hausner et al. [56] and Rabaan et al. [57], who documented the risks of consuming raw dairy products in rural settings. Our findings are consistent with numerous studies across tropical and subtropical regions. In Southern Brazil, Antinarelli, Silva, Guimarães, Terror, Lima, Ishii, Muniz, and Coimbra [44] found a seroprevalence of 44.4%, with higher rates in rural areas where environmental exposure is significant. In Northeastern Brazil, risk factors such as poor water treatment and contact with cats contributed to a higher likelihood of infection [43].

The current study is in line with the general literature pointing to climate and sanitation conditions as important factors determining the dissemination of *T. gondii*. Studies by Dias, Lopes-Mori, Mitsuka-Bregano, Dias, Tokano, Reiche, Freire, and Navarro [54] and Antinarelli, Silva, Guimarães, Terror, Lima, Ishii, Muniz, and Coimbra [44] confirm that socioeconomic conditions and rural living are major contributors to parasite transmission in Brazil.

The results of this study suggest an urgent need for targeted public health interventions. Educational campaigns on safe food handling, hygiene, and prenatal screening are essential for high-risk populations. The implementation of toxoplasmosis screening programs in regions with elevated seroprevalence, like Araçatuba, is vital to reducing the incidence of congenital toxoplasmosis.

The BHUs play a pivotal role in Brazil’s public healthcare system, especially in the prevention and control of diseases such as toxoplasmosis. These health centers are the first point of contact for the population and are fundamental for prenatal care, which includes screening for infections such as *T. gondii*. In the context of this study, the BHUs in Araçatuba served as a key platform for identifying the 55.14% seroprevalence of toxoplasmosis in pregnant women, highlighting their importance in early detection and intervention.

The BHU system also plays a crucial role in educating women about the risks of toxoplasmosis and how to prevent it. Many women in this study had low levels of education, a factor closely associated with higher rates of infection [55]. BHUs can offer targeted educational interventions, emphasizing the importance of avoiding undercooked meat, practicing proper hand hygiene, and ensuring safe food preparation—strategies proven effective in reducing infection rates [58].

Access to preventive care services, such as those provided by the BHUs, is important for reducing the prevalence of toxoplasmosis, particularly in lower-income populations. This study found that women with fewer years of formal education were more likely to be infected, a trend seen in other parts of Brazil [58]. BHUs, being community-based, are often more accessible to these populations and therefore play a key role in ensuring that preventive education and screening reach at-risk groups. The integration of comprehensive prenatal care at these health units ensures that even those with limited access to healthcare receive essential services, including regular serological monitoring for toxoplasmosis throughout pregnancy [59].

## 5. Conclusions

The study highlights a seroprevalence of 55.14% for *Toxoplasma gondii* among pregnant women in Araçatuba, São Paulo, which is consistent with findings from other regions of Brazil. It underscores the significant association of advanced age, multiparity, and lower educational levels with increased risk of infection. The results emphasize the importance of targeted public health interventions, such as prenatal screening and educational campaigns on safe food handling and hygiene practices, particularly for high-risk populations. Basic Health Units are essential for early detection and prevention and, furthermore, in improving access to healthcare and educational resources for vulnerable communities. Overall, these findings advocate for enhanced public health strategies to reduce the prevalence of congenital toxoplasmosis and its associated risks in endemic regions.

## Figures and Tables

**Figure 1 microorganisms-12-02183-f001:**
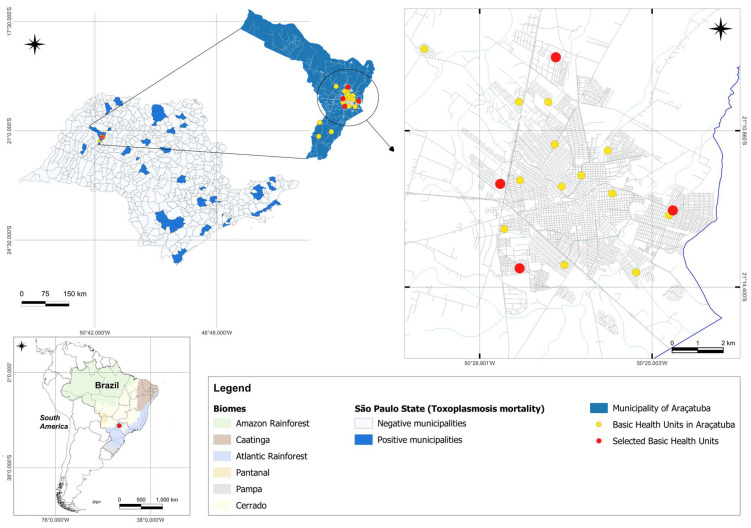
Map of the distribution of toxoplasmosis mortality in São Paulo State and selected Basic Health Units in Araçatuba, SP, Brazil.

**Table 1 microorganisms-12-02183-t001:** Demographic characteristics of pregnant women from Araçatuba, SP, by age group, marital status, pregnancies, and level of education.

Individual Risk Factors	Proportion (n)	CI95% ^1^	Standard Error
*Toxoplasma gondii*	55.14% (236)	50.38–59.81	2.40
Age range (Mean: 26.50; CI95%: 25.86–27.13)			
15–24	38.32% (164)	33.81–43.02	2.41
25–35	46.03% (197)	41.34–50.79	2.35
36–45	15.65% (67)	12.50–19.42	1.76
Marital status			
Single	20.79% (89)	17.20–24.91	2.34
Married	64.02% (274)	59.34–68.44	1.96
Other	15.19% (65)	12.08–18.91	1.73
Pregnancies			
1	37.85% (162)	33.36–42.55	2.34
>1	62.15% (266)	57.44–66.63	2.34
Level of Education			
Complete Elementary	25.93% (111)	21.99–30.31	2.12
Complete Medium	65.65% (281)	61.01–70.01	2.30
Graduated	8.41% (36)	6.12–11.45	1.34

^1^ CI: confidence interval.

**Table 2 microorganisms-12-02183-t002:** Prevalence of *Toxoplasma gondii* in pregnant women from Araçatuba/SP, by BHU, age group, marital status, pregnancies.

BHU	Individual Risk Factors	IgG (*Toxoplama gondii*)		Total	*p* Value
		Non-Reagent % (n)	Reagent % (n)		
A	Age range (Mean: 27.13; CI95%: 25.37–28.88; N = 72 *)				
	15–24	47.62% (10)	52.38% (11)	28.21% (21)	1.00
	25–35	45.24% (19)	54.76% (23)	58.97% (42)	
	36–45	44.44% (4)	55.56% (5)	12.82% (9)	
	Marital status				
	Single	47.06% (8)	52.94% (9)	23.61% (17)	0.73
	Married	47.83% (22)	52.17% (24)	63.89% (46)	
	Other	33.33% (3)	66.67% (6)	12.50% (9)	
	Pregnancies				
	1	44.44% (8)	55.56% (10)	25.00% (18)	0.89
	>1	46.30% (25)	53.70% (29)	75.00% (54)	
B	Age range (Mean: 29.05; CI95%: 27.16–30.95; N = 94 *)				
	15–24	59.46% (22)	40.54% (15)	39.36% (37)	0.005
	25–35	33.33% (13)	66.67% (26)	41.49% (39)	
	36–45	16.67% (3)	83.33% (15)	19.15% (18)	
	Marital status				
	Single	63.16% (12)	36.84% (7)	20.21% (19)	0.04
	Married	31.75% (20)	68.25% (43)	67.02% (63)	
	Other	50.00% (6)	50.00% (6)	10.71% (12)	
	Pregnancies				
	1	61.36% (27)	38.64% (17)	46.80% (44)	< 0.001
	>1	22.00% (11)	78.00% (39)	53.19% (50)	
C	Age range (Mean: 25.79; CI95%: 24.23–27.36; N = 89 *)				
	15–24	31.82% (14)	68.18% (30)	49.44% (44)	0.37
	25–35	18.75% (6)	81.25% (26)	35.96% (32)	
	36–45	15.38% (2)	84.62% (11)	14.61% (13)	
	Marital status				
	Single	31.58% (6)	68.42% (13)	19.40% (19)	0.16
	Married	18.18% (10)	81.82% (45)	61.80% (55)	
	Other	40.00% (6)	60.00% (9)	16.85% (15)	
	Pregnancies				
	1	22.22% (8)	77.78% (28)	40.45% (36)	0.65
	>1	26.42% (14)	73.58% (39)	59.55% (53)	
D	Age range (Mean: 28.41; CI95%: 26.87–29.97; N = 173 *)				
	15–24	69.35% (43)	30.65% (19)	35.84% (62)	0.02
	25–35	54.76% (46)	45.24% (38)	48.55% (84)	
	36–45	37.04% (10)	62.96% (17)	15.61% (27)	
	Marital status				
	Single	70.59% (24)	29.41% (10)	19.65% (34)	0.21
	Married	54.55% (60)	45.45% (50)	63.58% (110)	
	Other	51.72% (15)	48.28% (14)	16.76% (29)	
	Pregnancies				
	1	73.44% (47)	26.56% (17)	36.99% (64)	0.001
	>1	47.71% (52)	52.29% (57)	63.01% (109)	

* Mean: arithmetic means; CI95%: confidence interval; N = number of samples.

**Table 3 microorganisms-12-02183-t003:** Univariate multi-level logistic regression of risk factors associated with anti-*Toxoplasma gondii* antibodies among pregnant women from Araçatuba, SP.

Risk Factors	Reagent IgG n/%	Total	OR (CI95%) ^1^	*p* ^2^
Age range				
15–24	75 (45.73%)	164 (38.32%)	1.93 (1.42 < OR < 2.64)	<0.001
25–35	113 (57.36%)	197 (46.03%)		
36–45	48 (71.64%)	67 (15.65%)		
15–24	75 (45.73%)	164 (38.32%)	1	0.04
25–35	113 (57.36%)	197 (46.03%)	1.60 (1.05 < OR < 2.42)	
15–24	75 (45.73%)	164 (38.32%)	1	0.0006
36–45	48 (71.64%)	67 (15.65%)	3.00 (1.62 < OR < 5.54)	
25–35	113 (57.36%)	197 (46.03%)	1	0.05
36–45	48 (71.64%)	67 (15.65%)	1.88 (1.03 < OR < 3.43)	
Marital status				
Single	39 (43.82%)	89 (20.79%)	1.54 (1.11 < OR < 2.13)	0.0089
Married	162 (59.12%)	274 (64.02%)		
Other	35 (53.85%)	65 (15.19%)		
Single	39 (43.82%)	89 (20.79%)	1	0.01
Married	162 (59.12%)	274 (64.02%)	1.85 (1.14 < OR < 3.01)	
Single	39 (43.82%)	89 (20.79%)	1	0.25
Other	35 (53.85%)	65 (15.19%)	1.50 (0.79 < OR < 2.85)	
Married	162 (59.12%)	274 (64.02%)	1.24 (0.72 < OR < 2.14)	0.53
Other	35 (53.85%)	65 (15.19%)	1	
Pregnancies				
1	72 (44.44%)	162 (37.85%)	1	0.001
>1	164 (61.65%)	266 (62.15%)	2.01 (1.35 < OR < 2.99)	
Undercooked meat				
Yes	69 (56.10%)	123 (28.74%)	1.06 (0.69 < OR < 1.61)	0.88
No	167 (54.75%)	305 (71.26%)	1	
Manipulate soil or sand				
Yes	36 (58.06%)	62 (14.49%)	1.15 (0.67 < OR < 1.98)	0.72
No	200 (54.64%)	366 (85.51%)	1	
Eat raw vegetables and greens				
No	78 (55.32%)	141 (32.94%)	1.01 (0.67 < OR < 1.52)	0.96
Yes	158 (55.05%)	287 (67.06%)	1	
Presence of a vegetable garden at home				
Yes	17 (62.96%)	27 (6.31%)	1.41 (0.63 < OR < 3.16)	0.52
No	219 (54.61%)	401 (93.69%)	1	
Owning a pet cat				
Yes	47 (61.04%)	77 (17.99%)	1.34 (0.81 < OR < 2.22)	0.31
No	189 (53.85%)	351 (82.01%)	1	
Ingestion of unpasteurized cow’s or goat’s milk				
Yes	41 (66.13%)	62 (14.49%)	1.71 (0.97 < OR < 3.01)	0.21
No	195 (53.28%)	366 (85.51%)	1	
Level of education				
Complete Elementary	66 (59.46%)	111 (25.93%)	1.47 (1.04 < OR < 2.08)	0.029
Complete Medium	158 (56.23%)	281 (65.65%)		
Graduated	12 (33.33%)	36 (8.41%)		
Complete Elementary	66 (59.46%)	111 (25.93%)	1.14 (0.73 < OR < 1.78)	0.64
Complete Medium	158 (56.23%)	281 (65.65%)	1	
Complete Elementary	66 (59.46%)	111 (25.93%)	2.93 (1.33 < OR < 6.46)	0.01
Graduated	12 (33.33%)	36 (8.41%)	1	
Complete Medium	158 (56.23%)	281 (65.65%)	2.57 (1.23 < OR < 5.34)	0.02
Graduated	12 (33.33%)	36 (8.41%)	1	
Minimum wage				
≤1	24 (64.86%)	37 (8.64%)	1.62 (1.00 < OR < 2.63)	0.049
1 > 2	198 (55.46%)	357 (83.41%)		
>3	14 (41.18%)	34 (7.94%)		
≤1	24 (64.86%)	37 (8.64%)	1	0.01
1 > 2	198 (55.46%)	357 (83.41%)	1.85 (1.14 < OR < 3.01)	
≤1	24 (64.86%)	37 (8.64%)	1	0.25
>3	14 (41.18%)	34 (7.94%)	1.50 (0.79 < OR < 2.85)	
1 > 2	198 (55.46%)	357 (83.41%)	1.24 (0.72 < OR < 2.14)	0.53
>3	14 (41.18%)	34 (7.94%)	1	

^1^ odds ratio; ^2^ *p* value.

**Table 4 microorganisms-12-02183-t004:** Mixed-effect logistic regression results for factors associated with toxoplasmosis positivity among pregnant women from Araçatuba, SP.

Variables	Adjusted OR ^1^	CI95%	SE ^2^	*p* Value
Age range (36–45)	2.82	1.38 < OR < 5.78	1.03	0.005
Age range (25–35)	1.60	0.97 < OR < 2.64	0.41	0.063
Marital status (Married)	1.13	0.69 < OR < 2.03	0.34	0.693
Marital status (Single)	0.71	0.35 < OR < 1.42	0.25	0.326
Pregnant	1.48	0.91 < OR < 2.39	0.36	0.112
Level of education (Complete Elementary)	2.58	1.07 < OR < 6.19	1.15	0.034
Level of education (Complete Medium)	2.59	1.17 < OR < 5.74	1.05	0.019
Minimum wage (≤1)	2.10	0.73 < OR < 6.05	1.13	0.171
Minimum wage (1 > 2)	1.37	0.63 < OR < 2.98	0.54	0.427

^1^ odds ratio; ^2^ standard error

## Data Availability

The raw data supporting the conclusions of this article will be made available by the authors on request.

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
