# Peer review of "Seroprevalence of Toxoplasma gondii and Associated Risk Factors in Pregnant Women in Araçatuba, São Paulo, Brazil: A Multi-Level Analysis"

_microorganisms, 2024, doi:10.3390/microorganisms12112183_

Round 1

Reviewer 1 Report

Comments and Suggestions for Authors

Line 40. Toxoplasma should be in italics

There are many commas in the manuscript, so it cannot be read fluently.

Congenital toxoplasmosis is a public health issue of global interest; however, this research does not demonstrate anything new. In Brazil, atypical or recombinant strains of T. gondii can be identified, so the study would gain importance if genotyping of the infection strains was carried out, in order toidentify atypical and recombinant strains.

Reviewer 2 Report

Comments and Suggestions for Authors

The manuscript is well written. There are a few comments related to English that can be addressed with a thorough read-through. 

1) Line 48 - clarify what drinking water means. Perhaps, "drinking unregulated water"

2) Line 65 - restate this sentence - 'seroprevalence" is missing.

3) Line 81 - please rewrite for clarity.

4) Table 3 - If the presence of a cat in the household was not a significant risk factor, how are these individuals getting exposed to T. gondii?

4) Line 85 - rewrite - it does not make sense.

5) Be consistent with your citations in lines 254, 264, and others. Please change to Hassamen et al. 

Perhaps, include some of the study's limitations, the authors should consider, the quality of construction of households, the presence of peridomestic animals in the household, and the type of IgG assay in all four clinics (was the same brand/kit used?).

Comments on the Quality of English Language

Minor edits

Reviewer 3 Report

Comments and Suggestions for Authors

This article present an statistical study of the different influence factors for T. gondii infection between the population of Araçatuba, leading to the conclusion of advanced age, multiparity, and lower educational levels are the major risks.

The article gives and appropiate context to the T. gondii infections, and gives an extra effort to situate the evaluated population in comparison with Brazil and South america regions to compare situations. The numer of references is enough, the statistical methods are appropiate to evaluate the collected data and the conclusions are supported by those data. In my opinion this article will be of interest only to a specific and reduced target, but I think that it is acceptable to publish in present form.

Round 2

Reviewer 1 Report

Comments and Suggestions for Authors

-